# Immunological and Structural Characterization of Titin Main Immunogenic Region; I110 Domain Is the Target of Titin Antibodies in Myasthenia Gravis

**DOI:** 10.3390/biomedicines11020449

**Published:** 2023-02-03

**Authors:** Christos Stergiou, Rhys Williams, Jennifer R. Fleming, Vasiliki Zouvelou, Elpinickie Ninou, Francesca Andreetta, Elena Rinaldi, Ornella Simoncini, Renato Mantegazza, Julius Bogomolovas, John Tzartos, Siegfried Labeit, Olga Mayans, Socrates Tzartos

**Affiliations:** 1Tzartos NeuroDiagnostics, 115 23 Athens, Greece; 2Department of Biology, University of Konstanz, 78457 Konstanz, Germany; 31st Neurology Department, Eginition Hospital, National and Kapodistrian University of Athens, 157 72 Athens, Greece; 4Fondazione I.R.C.C.S., Istituto Neurologico Carlo Besta, 20133 Milano, Italy; 5School of Medicine, University of California, La Jolla, San Diego, CA 92093, USA; 6School of Medicine, Attikon University Hospital, National and Kapodistrian University of Athens, 124 62 Athens, Greece; 7DZHK Partner Site Mannheim-Heidelberg, Medical Faculty Mannheim, University of Heidelberg, 68167 Mannheim, Germany; 8Myomedix GmbH, 69151 Neckargemuend, Germany; 9Hellenic Pasteur Institute, 115 21 Athens, Greece; 10Department of Pharmacy, University of Patras, 265 00 Patras, Greece

**Keywords:** titin, myasthenia gravis, thymoma, main immunogenic region, epitope mapping, titin atomic structure

## Abstract

Myasthenia gravis (MG) is an autoimmune disease caused by antibodies targeting the neuromuscular junction (NJ) of skeletal muscles. The major MG autoantigen is nicotinic acetylcholine receptor. Other autoantigens at the NJ include MuSK, LRP4 and agrin. Autoantibodies to the intra-sarcomeric striated muscle-specific gigantic protein titin, although not directed to the NJ, are invaluable biomarkers for thymoma and MG disease severity. Thymus and thymoma are critical in MG mechanisms and management. Titin autoantibodies bind to a 30 KDa titin segment, the main immunogenic region (MIR), consisting of an Ig-FnIII-FnIII 3-domain tandem, termed I109–I111. In this work, we further resolved the localization of titin epitope(s) to facilitate the development of more specific anti-titin diagnostics. For this, we expressed protein samples corresponding to 8 MIR and non-MIR titin fragments and tested 77 anti-titin sera for antibody binding using ELISA, competition experiments and Western blots. All anti-MIR antibodies were bound exclusively to the central MIR domain, I110, and to its containing titin segments. Most antibodies were bound also to SDS-denatured I110 on Western blots, suggesting that their epitope(s) are non-conformational. No significant difference was observed between thymoma and non-thymoma patients or between early- and late-onset MG. In addition, atomic 3D-structures of the MIR and its subcomponents were elucidated using X-ray crystallography. These immunological and structural data will allow further studies into the atomic determinants underlying titin-based autoimmunity, improved diagnostics and how to eventually treat titin autoimmunity associated co-morbidities.

## 1. Introduction

Myasthenia gravis (MG) is a relatively rare, chronic autoimmune disorder caused by antibodies that block neuromuscular signal transmission, resulting in rapid muscle fatigue and skeletal muscle weakness [1,2,3]. The most frequent (in >80% of MG patients) target of these antibodies is the skeletal muscle nicotinic acetylcholine receptor (nAChR) at the postsynaptic end of the neuromuscular junction [1,4,5]. However, other components of the neuromuscular junction have also been implicated, e.g., muscle specific kinase (MuSK) in approximately 6% of MG patients [6,7], low-density lipoprotein receptor-related protein 4 (LRP4) in approximately 2% of MG patients [8,9], agrin and collagen Q [5,10]. All these extracellular autoantigens are invaluable for MG diagnosis. In addition to the neuromuscular junction receptors and extracellular proteins, there are also intracellular proteins of the striated muscle that can be recognized as autoantigens by the MG immune system. These include titin, cortactin and ryanodine receptors [11,12,13,14]. Anti-titin antibodies are the most thoroughly studied among the intracellular MG autoantigens and, even though they are not expected to contribute to MG pathology, together with the antibodies to ryanodine receptors, they are valuable biomarkers for MG subgroup classification (mostly for the differentiation between MG with or without thymoma) and treatment optimization [11,12,15]. Anti-titin positive MG is also associated with myocarditis, rheumatoid arthritis and lupus erythematosus [16]. Thus, more research is required to determine if titin autoantibodies may also promote autoimmune pathologies.

Titin is a giant, elastic, sarcomeric protein with a molecular weight of more than 3800 kD [17]. It is the largest known protein in the human body, spanning half the sarcomere, from Z discs to M lines, in skeletal and cardiac muscle [18,19]. Titin consists primarily of two types of protein domains: fibronectin type III (FnIII) domains and immunoglobulin (Ig) domains [20] (ca 100 residues in length) that account for over 90% of its mass [17] (Figure 1A). Ig and FnIII domains are joined by short linker sequences that lead to the formation of poly-domain tandems along the length of the titin chain [21]. The FnIII and Ig modules of titin are highly conserved and display related folds hosting equivalent sequence motifs [22,23,24]. Furthermore, titin domains commonly form repeats along the chain reflecting the evolution of this protein by serial genetic duplication events, which results in a certain homogeneity of the titin chain in sequence and structure. Despite the repetitive composition of titin, a single main immunogenic region (MIR) has so far been identified in its chain that elicits the reactivity of almost all known anti-titin antibodies in MG patients [25]. This 30 kD segment, also termed MGT30, consists of a domain triplet of composition Ig-FnIII-FnIII located in the titin segment at the junction of the I- and A-bands (I/A) and corresponding to domains 4 to 6 in this region of the chain [25], i.e., tandem I109-I111 also termed I/A4-I/A6, (nomenclature as in Bang et al., 2001 [17]). The epitope for anti-titin antibodies within this tandem is unknown.

Notably, the titin MIR locus is recognized by anti-titin antibodies, found mostly in MG patients positive for antibodies against nAChR. Generally, the presence of anti-titin antibodies in MG patient sera is related to age and to the presence of thymoma [26,27,28]. The thymus is the prime site of autosensitization and autoimmunity to AChR in MG pathogenesis [29]. Approximately 70% of MG patients present thymic changes, including hyperplasia and thymoma; the latter can be found in about 20–25% of MG patients worldwide [30]. Thymoma, likely via the AChR it contains and the defective autoimmune regulator (AIRE) or HLA class II expression, is believed to be involved in the induction of MG in thymoma-MG. Being a tumor, thymoma presence must be identified for thymectomy to follow [31]. The antibody prevalence is very high (50–80%) in late-onset MG (i.e., MG onset after 40 years of age) independently of thymoma presence, and only 6% in early-onset MG (i.e., MG onset up to 40 years old) [1,11,15,26,32,33]. Specifically, for early-onset MG, the presence of anti-titin antibodies strongly indicates thymoma, as 50–95% of early-onset (<40–50 years old) MG patients with thymoma are anti-titin positive, and only a few non-thymomatous early-onset MG patients are found anti-titin positive [1,26]. In parallel, there is a correlation between anti-titin antibodies and a more severe MG manifestation [15,26,34,35]. Therefore, evidence demonstrates that anti-titin antibodies are a very useful biomarker for MG diagnosis and prognosis, especially for thymoma diagnosis.

In order to advance the understanding of MIR antigenicity and support the development of more sensitive detection protocols, it is of great value to further dissect the fine antigenic repertoire of the MIR locus in titin and search for possible correlations to various clinical characteristics, such as the presence of thymoma, the severity of the symptoms, the age at onset, and patient gender. In this study, we recombinantly expressed titin protein fragments spanning the MIR locus I109-I111 as well as various non-MIR domains to serve as controls and performed ELISA, competition assays and Western blotting experiments with MG patient sera. We identified the middle MIR domain, I110, as the sole location of the epitope(s) of anti-titin antibodies in MG, and possibly in other diseases. Further, we resolved the 3D structure of the MIR tandem in titin, revealing the structure of the I110 domain both in isolation and within its context in the poly-domain chain.

## 2. Materials and Methods

### 2.1. Sera, Population Sample and Clinical Features of Patients

Sera were obtained from MG patients (and healthy controls, HC) referred to Tzartos NeuroDiagnostics, Athens, and to Laboratorio Diagnostica Neuroimmunologica, I.R.C.C.S. Fondazione Istituto Neurologico “C. Besta” after written informed consent. The study was conducted in accordance with the Declaration of Helsinki and was approved by the Institutional Review Boards of the Attikon University Hospital (protocol code B’ NEUR. EBD 280/17.5.21; date of approval: 27 July 2021) and the Fondazione IRCCS Istituto Neurologico Carlo Besta, for studies involving humans. Informed consent was obtained from all subjects involved in the study. MG was diagnosed based on clinical features, electrophysiological and serological findings. Most patients in the study were nAChR and titin antibody positive. MG patients were of various MG subtypes, including patients with thymoma, with hyperplastic or involuted thymus, of early or late-onset MG, males and females. The participating teams provided information always in anonymity for the samples with respect to MG severity, clinical characteristics and thymic histology and pathology.

### 2.2. Recombinant Production οf Titin Protein Fragments

Titin components I99-I101 (Uniprot entry Q8WZ42; residues 13388–13655), MIR I109-I111 (residues 14319–14615), I110-I111 (residues 14415–14615) and isolated I110 (residues 14415–14509) and I111 (residues 14510–14615) domains were cloned into the expression vector pETM11 (EMBL vector collection) that adds an N-terminal His6-tag and a Tobacco Etch Virus (TEV) protease cleavage site prior to the inserted gene. All clones were confirmed by sequencing.

The expression of all protein products was in the *Escherichia coli* strain BL21 (DE3) Rosetta (Merck Millipore, Burlington, MA, USA) grown at 37 °C up to an OD600 of 0.6–0.8 in Luria-Bertani medium supplemented with 25 μg/mL kanamycin and 34 μg/mL chloramphenicol. Expression was induced with 0.5 mM isopropyl-β-D-thiogalactopyranoside (IPTG) and growth continued for 18 h at 18 °C. Bacterial pellets, harvested by centrifugation at 5000× *g* and 4 °C, were resuspended in lysis buffer (25 mM MOPS pH 8.0, 300 mM NaCl, 1 mM β-ME) supplemented with 20 µg/mL DNase I (Sigma Aldrich, St. Louis, MO, USA) and one complete EDTA free protease inhibitor cocktail tablet (Roche) per liter of cell culture. Lysis was by sonication on ice followed by clarification of lysate by centrifugation at 39,000× *g* for 45 min. Following centrifugation, the cell supernatant was syringe filtered (0.22 µm) and applied to a 5 mL HisTrap HP column (Cytiva, Marlborough, MA, USA) connected to an Åkta FPLC system (GE Healthcare, Chicago, IL, USA) equilibrated in lysis buffer containing 20 mM imidazole. Elution was by continuous imidazole gradient. Next, eluted samples were dialyzed overnight into 25 mM MOPS pH 8.0, 50 mM NaCl, 1 mM β-mercaptoethanol and, for His6-tag removal, incubated with TEV during dialysis. The resulting sample was subjected to subtractive Ni^2+^ affinity purification using a HiTrap HP column (Cytiva, Marlborough, MA, USA). Anion exchange chromatography was then performed using a 5 mL QFF column (Cytiva, Marlborough, MA, USA) equilibrated in 25 mM MOPS pH 8.0, 50 mM NaCl, 1 mM β-mercaptoethanol. Then, size exclusion chromatography was performed on a Superdex S75 16/60 HiLoad column (GE Healthcare, Chicago, IL, USA) in 25 mM MOPS pH 7.5, 50 mM NaCl, 1 mM DTT. The resulting samples were >98% pure based on SDS-PAGE. The samples were flash frozen in liquid nitrogen and maintained frozen at −80 °C until further use.

The production of M1-M2 [36] and the titin kinase region A170-Kin-M1 [37,38] have been previously reported. The production of A160-A170 (residues 23648–24731) was as in Bucher et al. [24].

### 2.3. Synthetic Peptides

In an effort to precisely localize the epitope(s) in the MIR segment of titin, nine peptides (eight 18-amino acids and the last one 20-amino acids long) covering domain I110 (100 amino acids) and overlapping by 8 amino acids were synthesized commercially by BioCat GmbH (Heidelberg, Germany). The specific peptides are shown in Figure 1B.

### 2.4. ELISA and Antibody Competition Assay

An in-house ELISA was used for the detection of anti-titin antibodies mostly as previously described [12]. In detail, 96-well plates (Immulon 2 HB) were coated with the titin recombinant fragments (or synthetic peptides) or BSA, in 0.01 M carbonate buffer pH 9.6 (0.5 μg peptide per well) for 90 min at RT. Then, they were washed 4 times with PBS-Tween 0.05% and once with PBS. The plates were blocked with 4% BSA in PBS for 1 h at RT, before being washed again as previously. Next, 100 μL of diluted sera (1/100 in 4% BSA in PBS) were added and incubated for 90 min at RT. Then, horseradish peroxidase-conjugated rabbit anti-human IgG (Dako), diluted 1/5000 in 4% BSA in PBS was added and incubated for 90 min, RT, followed by washes and 3,3′,5,5′-tetramethylbenzidine (TMB) substrate. The reaction was stopped with 0.3 M sulphuric acid after 10 min and color development was quantified at 450 nm.

For the antibody competition assays, sera (at dilution 1/100 in 4% BSA in PBS) were treated with 5 μg of a titin fragment for 90 min, followed by their incubation with the immobilized titin fragment in the ELISA plate for the subsequent ELISA procedure as above.

### 2.5. Detection of Antibody Binding to I110 Fragment by Western Blotting

Recombinant titin samples (treated in sample buffer 0.125 M Tris-HCl pH 6.8, 4% SDS, 20% *v/v* glycerol, 0.2 M DTT, 0,02% bromophenol blue, and heated at 95 °C for 5 min), 1 μg per lane, were electrophoresed on a 20% SDS-polyacrylamide gel and transferred onto polyvinylidene difluoride (PVDF) membranes. The membranes were then incubated for 1 h at 25 °C with 3% BSA in PBS, then for 2 h at 25 °C with anti-titin-positive sera or HC sera, diluted 100 times in 0.2% BSA in PBS. They were then washed and incubated for 90 min at 25 °C with a 1:5000 dilution of HRP-conjugated anti-human antibody, washed with PBS, and incubated with 3,3′–diaminobenzidine (DAB) (Thermo Scientific), NiCl_2_ and H_2_O_2_ to visualize antibody binding, according to the manufacturer’s procedure.

To determine the percentage of antibodies bound on the blotted polypeptide, the incubated “depleted” sera were then used in ELISA with immobilized I110 and compared with the binding of the untreated sera.

### 2.6. Crystallization

Crystals of the isolated domain I110 were grown from solutions containing 0.1 M Tris pH 8.5, 30% (*v*/*v*) PEG 300. Following initial crystallization screening in nanovolumes and to obtain large single crystals, optimization screens were set up using the hanging drop method in 48-well VDX plates (Hampton Research) at 19 °C. Drops consisted of 1.5 µL of protein solution at ∼15 mg/mL and 1.5 µL of reservoir solution, with reservoirs containing 100 µL of mother liquor.

Crystals of I110-I111 were obtained as described above from solutions consisting of 24% (*w*/*v*) PEG 1500, 20% (*v*/*v*) glycerol, while crystals of full-length MIR were grown from 0.1 M MES pH 6, 10% (*w*/*v*) PEG 6000. For X-ray data collections, crystals were harvested and vitrified in liquid nitrogen directly in native mother liquor, with the exception of full-length MIR where the mother liquor was supplemented with 35% ethylene glycol.

### 2.7. Crystallographic Structural Elucidation

For all crystals, X-ray diffraction data were collected at the Diamond Light Source synchrotron (Didcot, UK) on beamline I02 under cryo-conditions (100 K). In all cases, data processing used the XDS suite [39] and phasing was performed by molecular replacement in PHASER [40]. For the isolated domain I110, the crystal structure of fibronectin domain A170 from titin (extracted from PDB entry 2NZI; [41]) was used as a search model in molecular replacement. The structure of I110-I111 used the structure of I110 as a search model for both FnIII components, while the structure of MIR was phased using the individual domains I110 and I111 of the dual structure plus an AlphaFold model of Ig domain I109 (obtained from https://alphafold.ebi.ac.uk/download (accessed on 28 Nov 2022); entry AF-Q8WZ42-F71-model_v4_14001-154001). Manual model building was performed in COOT [41] and refinement in Phenix.refine [42] using isotropic B-factors and TLS refinement. Statistics for X-ray data processing and model refinement are given in Table 1.

## 3. Results

### 3.1. Patient Sera Showed Binding Exclusively against I110-Containing MIR Protein Fragments

The following titin protein fragments were immobilized on ELISA plates: full-length MIR (I109-I111) and its subcomponents I109-I110, I110-I111, I110 and I111 (during the course of this study, we did not succeed in the recombinant production of the isolated domain I109 in soluble form); and the titin fragments outside the MIR region: the triple Ig tandem I99-I101 from the distal I-band, the 11-domain tandem A160-A170 from titin’s A-band comprising 5 Ig and 6 FnIII, the multi-domain titin kinase region A170-Kin-M1 including the FnIII domain A170 and the Ig M1, and the Ig-duet M1-M2 from the M-line (Figure 1A). In total, fragments outside the MIR region accounted for 11 Ig and 7 FnIII domains as well as the unique titin kinase domain, providing test sets for representative components of I-band, A-band and M-line regions of titin.

We initially tested 10 sera from MG patients already found positive for anti-titin MIR antibodies and a HC serum. All MG sera were bound to all MIR fragments except for I111, while none of them were bound to any titin protein fragment outside the MIR region, despite these containing domains that share conservation with MIR components (Figure 2A). The only common element between all MIR protein fragments which were bound to sera in this ELISA screening is the central FnIII domain, I110, suggesting that this domain may contain the binding epitope(s) for anti-titin antibodies in MG patients. The average values of Figure 2B show more clearly that binding to I109-I110, I110-I111 or I110 alone does not significantly differ from their binding to the whole MIR, suggesting that the main binding interface is located in the common element, I110, and that the other domains play no or only a minor role in antibody binding.

### 3.2. Titin FnIII Domain I110 May Contain the Binding Epitope(s) for All/Most Anti-Titin Antibodies in MG Patients’ Sera

In order to determine whether domain I110 is indeed the primary or only immunogenic locus in MG patients, we performed competition assays with soluble I110 before the ELISA assay. Specifically, the sera of Figure 2 were pre-incubated with the soluble I110 domain, or the M1-M2 fragment as control, and then they were tested in ELISA assays against all antibody-binding MIR fragments, i.e., full-length MIR and its components I109-I110, I110-I111, I110. Interestingly, pre-incubation with domain I110 abolished binding to all MIR fragments to levels equivalent to negative controls for all the sera tested. On the contrary, pre-incubation with the negative control M1-M2 did not affect binding of any serum to any MIR component (Figure 3A). These data supported the conclusion that I110 constitutes the main antigenic locus in the MIR segment of titin.

For further confirmatory evidence, we then tested binding of additional 20 MG sera to the MIR tandem and its I110 domain component, and found that binding was practically identical for both samples. In addition, antibody binding to either sample was totally inhibited by preincubation of the 20 sera with the soluble I110 domain (Figure 3B). These results are in full agreement with the above analysis and confirm that I110 contains the primary MIR epitope, at least for the tested MG sera.

### 3.3. Correlation of Antigenic Epitopes with MG Clinical Characteristics

Most sera assayed above (Figure 2 and Figure 3) originated from MG patients with limited available clinical details. In order to assess whether the unique immunogenicity of I110 applies to all MG subgroups, and whether there is any correlation between antibody binding and MG characteristics, including thymic state of the patients, age of MG onset (early onset, <40 years old, or late onset, >40 years old) and patients’ gender, we proceeded to test sera from additional 32 MG patients of known, heterogeneous characteristics. Figure 4 shows antibody binding tests for the nine titin samples (described above) that used the extra 32 sera from MG patients were divided into three groups: a. patients with hyperplastic thymus (mostly with early onset MG, 6/7 patients; all females), b. patients with involuted thymus (the majority with late onset MG, 9/14 patients, mostly females), and c. patients with thymoma (the majority with early onset MG, 8/11 patients; mostly males). Although the average binding to I110 was marginally lower than that to the other three binding samples (MIR, I109-I110 and I110-I111) (Figure 4D), the differences were not significant. Only the serum from one patient in the whole study (the encircled thymoma patient six in Figure 4C), was bound somewhat more poorly to I110 (ca 47%) than to the other three binding samples. Nevertheless, none of the 32 sera (as well as the 10 sera of Figure 2A) were bound to any sample that did not contain the I110 domain.

### 3.4. Titin Antibodies of Sera from Non-MG Patients Also Seem to Bind Exclusively to the I110 MIR Domain

We then tested a few available anti-titin sera from non-MG patients. Sera from patients suspected for autoimmune encephalitis or paraneoplastic syndrome are routinely tested for the presence of antibodies to a panel of paraneoplastic antigens. Sera from 15 patients suspected for autoimmune encephalitis or paraneoplastic syndrome were earlier screened for binding to a commercial panel of 12 paraneoplastic antigens, including titin MIR, by dot blot (Euroimmun, Lubeck, Germany). We tested the binding of these 15 sera by ELISA to titin MIR and to its domains I110 and I111. Figure 5 shows that there is a good correlation between dot blot anti-MIR intensity and ELISA anti-MIR values, however only the strongly positive by dot blot sera (dot intensity > 45 units) were positive by the ELISA for the same full-length MIR sample. In addition, serum no. 4, although strongly positive by dot blot, was negative by ELISA. Importantly, as with the MG sera, binding to domain I110 was very similar to that observed against the full-length MIR tandem, whereas, similarly to the MG sera, practically no serum was bound to domain I111.

### 3.5. Antibodies Can Bind to the SDS-Denatured I110 Domain

In order to further investigate whether the epitope(s) on I110 are three-dimensional or sequential, we performed SDS-PAGE Western blots, where the SDS-denatured DTT-reduced titin samples (I109-I110, I110, I110-I111 and I111) were subjected to SDS-electrophoresis and blotted on polyvinylidene difluoride (PVDF) membranes. Then, the membranes were incubated with MG anti-MIR sera (and with HC sera). Figure 6A shows that all 3 anti-I110 positive by ELISA sera were bound also to the SDS-denatured I109-I110, I110-I111 and I110 samples by Western blot (i.e., all those which contain the I110 domain) but not to the I111 sample. In contrast, the HC serum did not show any binding to any of the samples. In another two Western blots, four more anti-titin sera and two HC were also tested with the same result. These results suggested that the antibody epitope in I110 is formed by a linear sequence motif and is not the result of a three-dimensional, spatial arrangement of amino acid residues with specific conformation.

Because Western blot staining does not show whether the bound antibodies represent a major or a minor fraction of the serum’s antibodies to the antigen and binding of a minor antibody fraction could be non-representative, we then measured quantitatively the fraction of antibodies binding to the blotted I110 sample. Specifically, after incubation of another five anti-I110 sera with the blotted I110 (Figure 6B), we subjected the “depleted” sera to ELISA with immobilized I110 and compared their binding to that of the untreated (non-depleted) sera. Figure 6C shows that all five sera were depleted to various degrees, with an average depletion 67%; i.e., the majority of the antibodies were bound to the SDS-denatured I110.

### 3.6. The Use of Synthetic Peptides from the I110 Domain Could Not Identify the Exact Antibody Epitope(s)

Since MG patient sera could bind to the denatured I110 domain, we aimed to identify the specific linear epitope(s) in I110. For this, we designed and obtained nine synthetic peptides (the eight 18-amino acids, the ninth 20-amino acids in length), covering the full I110 domain (100 amino acids) and overlapping by eight amino acids (Figure 1B). We tested binding of four anti-I110 sera to the immobilized peptides using ELISA; none were bound significantly to any peptide. Then, we tested the peptides in competition assays, i.e., the ability of the soluble peptides to bind antibodies in the sera and thereby inhibit antibody binding to the immobilized MIR. Here, sera were preincubated with the peptides, individually and in groups (three groups of three peptides each and two groups of four and five peptides each), and the mixtures were incubated with the ELISA-immobilized MIR; their binding was compared to that of the untreated sera. No inhibition of antibody binding was observed. Therefore, linear epitope(s) in I110 could not be identified.

### 3.7. Crystal Structure of Titin MIR and Its Subcomponents

To gain an insight into the possible molecular determinants of I110 recognition by anti-titin antibodies, we elucidated the 3D-structure of the isolated I110 domain, the double tandem I110-I111 and the full-length MIR region I109-I110 using X-ray crystallography (Figure 7A). The structures showed that the domains contain structure and sequence features typical of Ig [42] and FnIII [22,24] constituents of titin, and that share high levels of sequence and structural conservation with other domains of titin. Domain I110 also presented such typical features without detectable deviations (Figure 7B). In fact, its structural comparison to FnIII domains from titin with known 3D-structure revealed a remarkable agreement: namely, RMSD_Cα_ values for superimpositions onto domains A77, A78 and A170 yielded values of 0.81 Å, 1.02 Å and 1.18 Å, respectively. The I110 domain showed a stable and same structure when isolated and arrayed with its neighboring domains. I110 is joined to the preceding Ig through a short two-residue linker sequence, residues AM, with the relative orientation of both domains being fixed by two interdomain salt bridges (Figure 7A). C-terminally, I110 is joined to the subsequent domain by a flexible three residue long, hydrophilic linker, STA. No direct contacts exist between I110 and I111, so that the linker sequence acts as an effective hinge permitting the free repositioning of both domains. Accordingly, the structures of I110-I111 and I109-I111 show a different mutual arrangement of these domains. In brief, structural analysis did not lead to the identification of unique features in I110 that might support its unique recognition by anti-titin antibodies. An analysis of sequence conservation relative to FnIII used as negative controls in this study reveals a subset of surface residues unique to I110, that could mediate its specific recognition by anti-titin antibodies (Figure 7B). The available crystal structures will support future studies that could resolve the role of these candidate residues in I110′s antigenicity and permit mapping their formation of epitope(s).

## 4. Discussion

Titin antibodies are an important biomarker for the presence of thymoma in MG, i.e., for the instruction of thymectomy operation, as well as for diagnosis and prognosis of severe MG disease [1,15,26]. Earlier pioneering studies revealed that out of the ~3800 KDa of the gigantic titin protein, only a segment corresponding to <1% of its sequence is the sole or main target of the titin antibodies, named MIR (main immunogenic region) [25]. Yet, the MIR is still of considerable size, about 300 amino acids long (~30 KD) and consists of three domains. In the present study, we clearly showed that the single antigenic region of the titin MIR is its middle FnIII domain, I110.

We firstly showed that all 10 tested MG anti-MIR sera were bound to three of the four protein samples corresponding to MIR components, at least equally efficiently to their binding of the intact full-length MIR. In contrast, no serum was bound to I111 or to any of the four non-MIR samples tested; their OD_450_ was practically as low as that of the healthy controls. In fact, the average binding values of the three positive samples were marginally higher, though not significantly higher, than those for the full-length MIR. The common domain among the three binding samples is I110. Although we could not express the Ig domain I109 alone, the practically identical serum binding to I110 alone with that of their binding to the two larger samples containing I110 (I109-I110 and I110-I111), and to the full-length MIR, suggests that all or almost all anti-MIR antibodies of all 10 sera bind to domain I110.

Furthermore, in order to verify that I110 is indeed the only immunogenic MIR domain for MG patients, we performed competition assays in which the soluble I110 domain was firstly allowed to block the anti-I110 antibodies in the test sera before their incubation with the immobilized samples. We showed that the I110-treated sera had no remaining antibodies binding to any immobilized samples, whereas no effect was observed after serum pretreatment with the non-MIR fragment corresponding to M1-M2. This observation confirms that the I110 domain is the only target of all anti-MIR antibodies, at least in all tested 30 MG sera in Figure 3.

As described in the introduction, the presence of anti-titin antibodies in MG patients is heavily related to the thymic stage and age at onset of MG. This suggests that different mechanisms could cause expression of anti-titin antibodies in the different MG subgroups and, therefore, that the corresponding antibodies may differ between the different groups. The large majority of the 30 sera in Figure 2 and Figure 3 were from patients with limited available clinical data. In order to investigate whether anti-titin antibodies differ between MG subgroups, we then compared the titin binding pattern of sera from patients with hyperplastic thymus (most of early onset MG, <40 years old), with involuted thymus (the majority were of late onset MG, >40 years old), or with thymoma (the majority were of early onset MG). In all cases but one, the binding pattern to the nine samples assayed was nearly identical in all sera; I110 was again the necessary domain for antibody binding in all cases. Only in one case with thymoma, binding to I110 amounted to only 41% of the binding to the other fragments. The lower binding efficiency of these antibodies to the I110 domain could not be attributed to partial binding of this patient’s antibodies to non-I110 domains. This is because domain I111 did not bind, whereas both fragments containing I110 (I109-I110 and I110-I111) exhibited the maximum binding, suggesting also that I109 by itself was not bound to the remaining antibodies. Since the crystal structures of MIR domain tandems and isolated I110 (Figure 7) show that the structure of I110 is not affected by flanking domains, the apparent reduced binding could be simply due to the specific methodology used. For example, ELISA immobilization of the isolated I110 domain could have resulted in the masking or distortion of a specific epitope in all I110 molecules, whereas only a fraction of this epitope could be masked with the larger MIR fragments.

Although titin antibodies seem to be most frequent in MG patients, such antibodies have also been described in patients with paraneoplastic or encephalitis syndromes [43,44]. Indeed, some commercial dot blot diagnostic panels with several paraneoplastic antigens for the serum detection of corresponding antibodies also contain the titin MIR. To investigate whether the target domain of the titin antibodies of these non-MG patients differ from those detected in the MG patients, we tested for binding to the MIR and its I110 and I111 domains, 15 sera from patients with a suspect of paraneoplastic or encephalitis syndromes, positive for anti-titin MIR antibodies by dot blot. All six sera, which were bound significantly to full-length MIR by ELISA, were bound practically equally well to the I110 domain, but no binding was observed to the I111 domain. Therefore, it seems that independently of the autoimmune mechanism which induces the anti-titin antibodies, the epitope(s) are always located in the I110 domain.

In order to confirm the ELISA and competition results by an alternative approach, but also to determine whether the titin epitope(s) are conformational or sequential, we then tested antibody binding by Western blot approaches. Although in a single patient (patient no. six in Figure 4C) binding of ~60% of her anti-titin antibodies seem to be conformation-dependent, the conformational requirements of the anti-titin antibodies in all other tested sera were unclear. Such a knowledge would be important to instruct approaches for fine mapping of the titin epitopes. We therefore performed SDS-PAGE Western blots, in which the SDS-denatured MIR samples (I109-I110, I110-I111, I110 and I111) run under SDS-electrophoresis and blotted with anti-titin sera. The results of this alternative methodology confirmed that anti-MIR MG patient sera indeed bind only to the I110 domain (they were bound only to the I110 containing samples) and, importantly, suggested that the epitope(s) are sequential and not strictly conformation dependent. In fact, we found that at least a major fraction of the I110 antibodies (about 67%) can bind to the denatured I110.

Based on the observation that I110 antibodies bind to the SDS-denatured fragment, we then obtained and tested nine overlapping, synthetic, 18-residue long peptides that covered the full-length I110 domain. We used direct ELISA and competition ELISA, but unfortunately, we could not detect binding of the MG antibodies to any of the overlapping synthetic peptides. Although we do not know the reason for the complete absence of antibody binding, at least two possibilities could apply: a. attachment of the small synthetic peptides onto the ELISA plates might decrease their accessibility and mask the epitope(s); b. the relatively long synthetic peptides may not be in extended, unfolded conformation, but they may form non-native structures that do not correspond to their structural state within the context of the I110 domain. Unfortunately, structural analysis in this study did not lead to the identification of unique features in I110 that might explain its unique recognition by anti-titin antibodies. However, an analysis of differential sequence conservation has revealed a subset of surface residues unique to I110 that could mediate its specific recognition by anti-titin antibodies (Figure 7B). Thus, based on crystal structures and sequence analyses, the study of non-conserved residues in I110 in alternative peptides and the use of alternative antibody binding approaches might result in the localization of the exact titin epitope(s) within I110. As described already, anti-titin antibodies are found in early onset MG with thymoma and in late-onset MG independent of thymoma presence [1,26]; therefore, so far, measurement of anti-titin antibodies is useful only for early-onset MG patients. If fine epitope localization identifies the thymoma-specific epitopes and antibodies, it will be invaluable for thymoma diagnosis in all MG age groups.

In conclusion, in this paper we showed with three approaches that all titin antibodies, already known to bind to the titin MIR region, in fact bind on epitope(s) exclusively located in the middle MIR domain, the 97 amino acids long I110 domain. This applies independently of the groups of patients with titin antibodies, including MG patients with thymoma, hyperplastic or involuted thymus, early or late onset of the disease (which may follow different pathogenic mechanisms), or even non-MG patients. The exact titin epitope(s) could not be identified, but the data acquired indicate that they are mostly sequential and not strictly conformation dependent. The identified much smaller titin antigenic region should lead to easier antigen production and thus less expensive diagnostics, to lower non-specific binding, i.e., more specific diagnostics, and should facilitate research on the relation of anti-titin antibodies with thymoma and with MG induction. The 3D structure of the titin MIR will help future studies on the pathomechamisms acting in titin-based autoimmunity and should also improve diagnostics for titin antibodies as an important biomarker for thymoma and MG severity diagnosis.

## Figures and Tables

**Figure 1 biomedicines-11-00449-f001:**
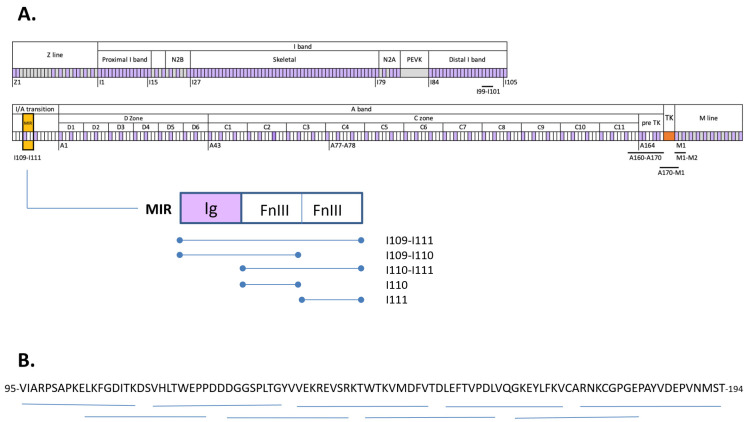
Schematic representation of the domain composition of titin and fragments used in the present study. (**A**) Domain composition of titin. Fibronectin domains are shown in white, Ig domains in purple and unique sequences in grey. Titin kinase (TK) is in orange. Titin sarcomeric regions and A-band super-repeats are annotated. The first domain of each zone is numbered. Segments studied in this work are underlined. The MIR segment and its constituents studied in this work are shown; (**B**) Amino acid sequence of I110 and its corresponding 9 synthetic peptides in this work (horizontal thin lines under the sequence).

**Figure 2 biomedicines-11-00449-f002:**
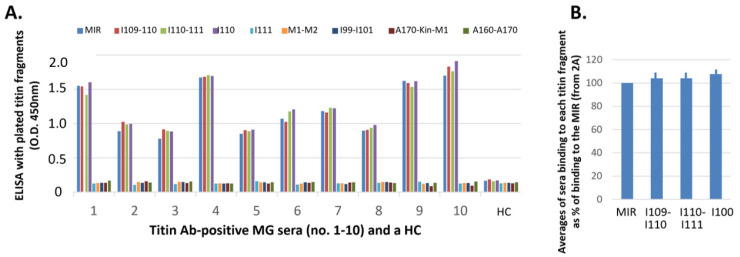
Binding of MG sera to the titin fragments by ELISA. Sera (from 10 MG patients and a healthy control, HC) were incubated with immobilized fragments: whole MIR, its domains I109-I110, I110-I111, I110 and I111, and the titin fragments outside the MIR region: M1-M2, I99-I101, titin kinase region A170-Kin-M1, and A160-A170. (**A**) It is shown that all sera were bound to all 3 fragments which contain the I110 domain, practically equally well with their binding to the whole MIR, but not to any other fragment. (**B**) Average percentages of antibody binding of the 10 sera to I109-I110, I110-I111, and I110, versus the values of antibody binding to the MIR. The differences are not statistically significant.

**Figure 3 biomedicines-11-00449-f003:**
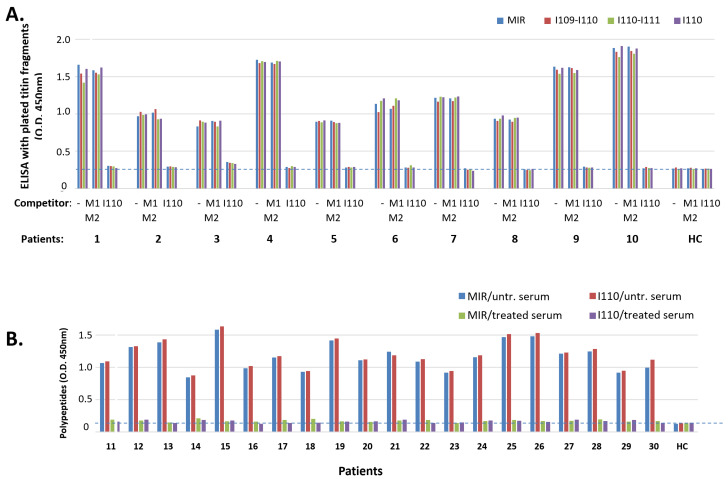
Efficiency of the I110 domain to block sera from binding to the MIR and MIR fragments. (**A**) Binding of the 10 MG sera of Figure 2 to the four MIR fragments and inhibition of binding by the soluble M1-M2 and I110 fragments. Each serum, before incubation with the immobilized titin fragment (MIR, I109-I110, I110 and I110-I111), was preincubated with the soluble M1-M2 or I110 fragments or with plain buffer (untreated (-)). It is shown that preincubation with I110 blocked all antibodies to all 4 immobilized fragments (all contain I110 domain), whereas the soluble M1-M2 fragment did not affect antibody binding to any of the immobilized fragments. (**B**) Binding of another 20 MG sera (no 11–30) to the MIR and I110 fragments (1st and 2nd columns of each patient), and its inhibition by the soluble I110 fragment (3rd and 4th columns). These data confirm that all anti-MIR antibodies bind to I110 epitopes.

**Figure 4 biomedicines-11-00449-f004:**
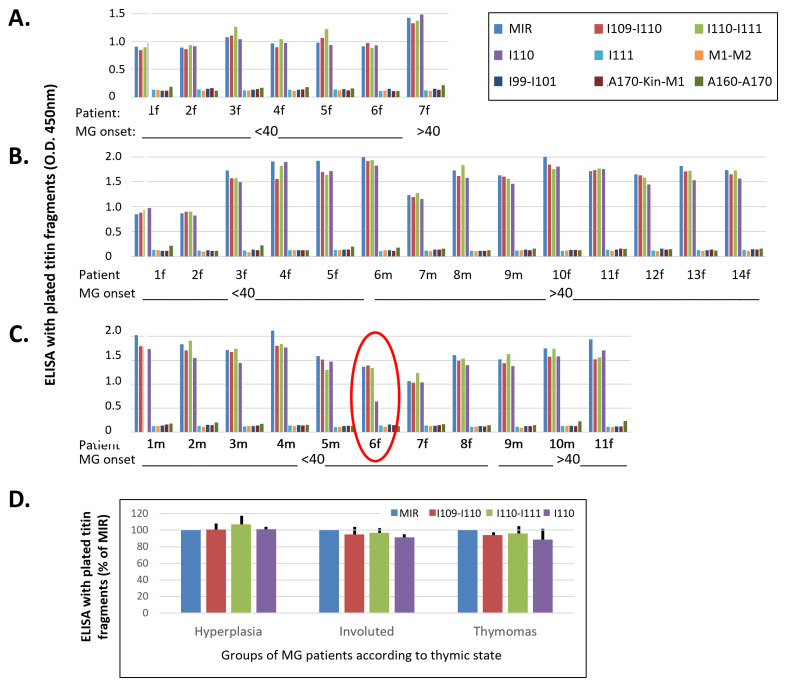
Search for correlation of thymoma state, age at onset and gender of MG patients versus the binding pattern to titin fragments. (**A**–**C**): Sera from 3 groups of MG patients with hyperplastic thymus (**A**), involuted thymus (**B**), or thymoma (**C**), further placed in subgroups of early onset (<40 years) and late onset MG, males or females, were tested for binding to the 9 fragments: MIR, I109-I110, I110-I111, I110, I111, M1-M2, I99-I101, titin kinase region A170-Kin-M1 and polyFn. In each group, first are shown the sera from early onset (<40) MG and then from late onset, and in each subgroup the male patients are placed first, followed by the females (marked with m, f next to patient number). The encircled patient C6 is the only one in the whole study whose serum was bound to I110 only about 47% of its binding to each of the other 3 samples. (**D**) Average percentages of antibody binding of the 3 groups of sera to I109-I110, I110-I111 and I110, versus the values of antibody binding to the MIR. The differences are not statistically significant.

**Figure 5 biomedicines-11-00449-f005:**
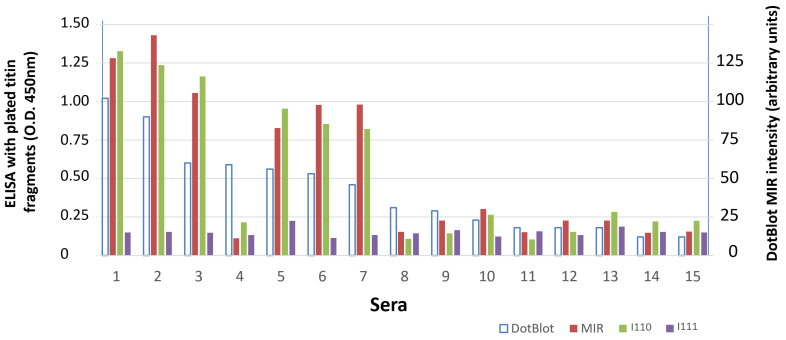
Binding to titin fragments of sera from patients suspected for paraneoplastic syndromes. Patients’ sera were initially screened for antibodies to 13 paraneoplastic antigens, including titin MIR, by dot blot. All sera which were found positive for titin MIR by dot blot (1st column for each serum shows the intensity of the dot in arbitrary units; right *Y*-axis) were further tested by ELISA for binding to the 3 MIR fragments: whole MIR, I110 and I111. The 3 solid bars for each patient represent ELISA values for the immobilized fragments: whole MIR, I110 and I111 (left *Y*-axis). It is shown that a) there is a good correlation between dot blot MIR intensity and ELISA MIR values, with only one exception (patient 4); b) as with the MG sera, binding to the I110 is very similar with that to the MIR, whereas practically no serum was bound to I111.

**Figure 6 biomedicines-11-00449-f006:**
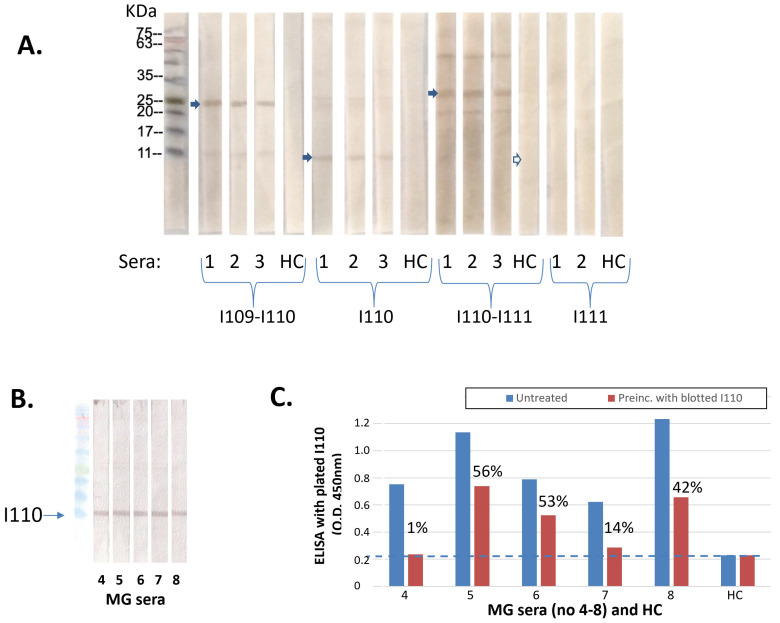
Binding of anti-titin antibodies to denatured I110 domain by Western blotting. (**A**) Titin MIR samples were SDS-electrophoresed and transferred onto polyvinylidene difluoride (PVDF) membranes, which were then incubated with 1/100 dilutions of 3 test MG sera, and a HC serum. It is shown that all MG sera recognized the I110 and the two I110-containing fragments but not the I111 fragment, whereas the HC serum showed no binding to any fragment. (**B**) Five more MG anti-MIR/I110 sera were incubated with blotted I110. (**C**) The incubated “depleted” sera of 6B were then used in ELISA with immobilized I110 fragment and compared with the binding of the corresponding untreated sera. Blue columns: untreated sera; red columns: treated “depleted” sera. Percentages above red columns denote the % of unbound I110 antibodies (after deduction of the background values marked with the dotted line). The average % of unbound antibodies was 33%.

**Figure 7 biomedicines-11-00449-f007:**
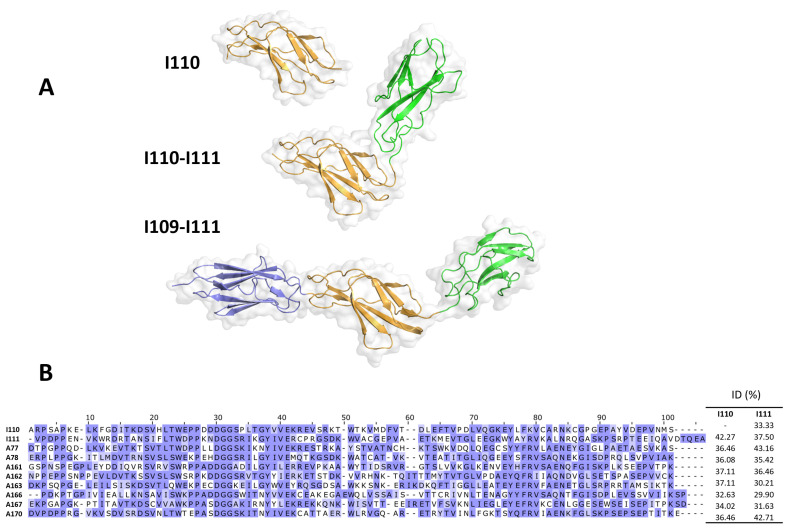
Crystal structures of the MIR I109-I111 tandem and its subcomponents. (**A**) Crystal structures where individual domains are color-coded across structures. All structures have been positioned as to agree in their orientation of the I110 domain (orange); (**B**) Conservation across I110 and fibronectin domains are used as negative controls in this study. Domains A77-A78 of known crystal structure are included as used in structural comparison. Conservation is color-coded according to BLOSUM62 and sequence identities across domains is provided (right). Non-conserved residues in I110 might form part of a potential epitope site.

**Table 1 biomedicines-11-00449-t001:** X-ray diffraction data and model refinement statistics.

	I110	I110-I111	I109-I111 (MIR)
PDB code	8BW6	8BVO	8BXR
Space group	*P*4_1_2_1_2	*P*2_1_	*P*2_1_2_1_2_1_
Cell dimensions			
a,b,c (Å)	45.86, 45.86, 112.02	34.28, 46.26, 71.34	30.36, 89.11, 128.05
α,β,γ (°)	90, 90, 90	90, 100.73, 90	90, 90, 90
Copies in ASU	1	1	1
**Data Processing**			
Beamline	I02 (Diamond)	I02 (Diamond)	I02 (Diamond)
Detector	PILATUS 6M	PILATUS 6M	PILATUS 6M
Wavelength (Å)	0.9795	0.9795	0.9700
Resolution (Å)	42.4–1.95	30.0–2.55	30.0–2.7
	(2.00–1.95) ^a^	(2.60–2.55) ^a^	(2.8–2.7) ^a^
No. Reflections	9305 (651)	7239 (401)	8696 (893)
R_sym_(I) (%)	4.6 (240.3)	16.0 (152.6)	19.3 (178.5)
	23.09 (0.93)	6.04 (1.13)	8.32 (1.18)
CC1/2 (%)	100.0 (52.1)	98.2 (42.4)	99.2 (51.5)
Completeness (%)	100.0 (100.0)	98.9 (100)	85.3 (87.6)
Multiplicity	12.36 (12.43)	3.19 (3.32)	8.04 (8.19)
**Model Refinement**			
No. working/free			
Reflections	8834/466	6715/509	8235/435
Rwork/Rfree (%)	21.74/25.39	22.35/28.72	22.54/27.14
No. Atoms Protein	805	1559	2326
No. Atoms Solvent	61 ^b^	44	61 ^c^
R.m.s.d.:			
Bond length (Å)	0.008	0.008	0.003
Angles (°)	0.975	1.016	0.532
Ramachandran plot			
Favored (%)	91.75	91.33	94.59
Disallowed (%)	0	0	0

^a^ values in parentheses correspond to the highest resolution shell; ^b^ Solvent included 7 × ethylene glycol, 1 × polyethylene glycol and 1 × calcium ion; ^c^ Solvent included 6 × ethylene glycol.

## Data Availability

Crystal structure coordinates and experimental structure factors have been deposited with the Protein Data Bank (entries 8BW6, 8BVO and 8BXR for I110, I110-I111 and full-length MIR, respectively). The corresponding X-ray diffraction images have been deposited with Zenodo with accession codes 0.5281/zenodo.7428754, 10.5281/zenodo.7428686, and 10.5281/zenodo.7428617, respectively. The remaining data presented in this study are available upon request from the corresponding authors.

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
