# Peer review of "Immunological and Structural Characterization of Titin Main Immunogenic Region; I110 Domain Is the Target of Titin Antibodies in Myasthenia Gravis"

_biomedicines, 2023, doi:10.3390/biomedicines11020449_

Round 1

Reviewer 1 Report

1. This paper shows that anti-titin antibodies in myasthenia gravis bind exclusively to titin's middle MIR domain, I110.
2. A major concern with this paper is the incorrect grammar in the use of the word "bound." throughout the paper Please see the comments in the attached pdf.
3. Also, Line 60, identification of the intracellular proteins cited in references 11-14 is needed.
4. The methodology and procedures for antibody binding are presented in rigorous detail, enabling replication of the study to verify the study findings.
5. The authors' findings apply to all myasthenia gravis (MG) patients in the study, including controls without MG.
6. The authors suggested that future studies use alternative antibody binding approaches to locate specific titin epitope(s) within I110, the binding site(s) of the antibodies.

Author Response

Response to Reviewer 1 Comments

Point 1: This paper shows that anti-titin antibodies in myasthenia gravis bind exclusively to titin's middle MIR domain, I110.

Response 1: We agree with the reviewer.

Point 2: A major concern with this paper is the incorrect grammar in the use of the word "bound." throughout the paper Please see the comments in the attached pdf.

Response 2: We have incorporated all the corrections.

Point 3: Also, Line 60, identification of the intracellular proteins cited in references 11-14 is needed.

Response 3: It has been corrected (now line 64).

Point    4: The methodology and procedures for antibody binding are presented in rigorous detail, enabling replication of the study to verify the study findings.

Response 4: We thank the reviewer for the positive comment

Point 5: The authors' findings apply to all myasthenia gravis (MG) patients in the study, including controls without MG.

Response 5: We thank the reviewer for the positive comment

Point 6: The authors suggested that future studies use alternative antibody binding approaches to locate specific titin epitope(s) within I110, the binding site(s) of the antibodies.

Response 6: We agree with the reviewer.        

Reviewer 2 Report

The manuscript by Stergiou et al. provides a characterization of the main immunological region (MIR) in the contractile machinery protein Titin. Autoantibody directed against Titin are markers of thymoma and the severity of myasthenia gravis (MG). Therefore, detailed characterization of the primary antibody binding site on the Titin sequence could contribute to a better understanding of the pathological mechanisms and provide new diagnostic tools. The authors prove that the I110 fragment within the MIR is the main site to bind antibodies from MG patient sera. The manuscript is well written, experiments are carefully planned, and the results are clear. I think this work should be published.

Author Response

Response to Reviewer 2 Comments

Point 1: The manuscript by Stergiou et al. provides a characterization of the main immunological region (MIR) in the contractile machinery protein Titin. Autoantibody directed against Titin are markers of thymoma and the severity of myasthenia gravis (MG). Therefore, detailed characterization of the primary antibody binding site on the Titin sequence could contribute to a better understanding of the pathological mechanisms and provide new diagnostic tools. The authors prove that the I110 fragment within the MIR is the main site to bind antibodies from MG patient sera. The manuscript is well written, experiments are carefully planned, and the results are clear. I think this work should be published.

Response 1: We thank the reviewer for the positive comments

Reviewer 3 Report

The group of  Stergiou et al. investigated the antigenic repertoire of the MIR locus in titin and search for possible correlations to various clinical characteristics. The manuscript shows new results but is not easy to understand. I have the following comments: First of all, the title is very long, contains abbreviations and is difficult to understand. Can't it be summarised more concisely and, above all, more briefly? The authors note that anti-titin antibodies play a role in the diagnosis of MG. Does this also apply to the other autoantibodies? It is not quite clear to me what the advantage is of clarifying the titin epitopes. In the introduction, the authors should also mention the connection between MG and thymoma. This is not clear and is certainly not obvious to every reader. It is not clear to me how one wants to further characterise the MG on the basis of the anti-titin-AK if there are no real differences between the individual sera. This rather speaks against the search for anti-titin-AK as a biomarker.

Author Response

Response to Reviewer 3 Comments

Point 1: The group of Stergiou et al. investigated the antigenic repertoire of the MIR locus in titin and search for possible correlations to various clinical characteristics. The manuscript shows new results but is not easy to understand.

Response 1: We believe the replies below, incorporated in the text, as well as those to reviewer 1, have made the manuscript satisfactorily easy to understand.

Point 2: First of all, the title is very long, contains abbreviations and is difficult to understand. Can't it be summarised more concisely and, above all, more briefly?

Response 2: The original title was: Immunological and Structural Characterization of the Titin MIR Locus; Identification of the Middle MIR Domain, I110, as the Main Target of Titin Antibodies in Myasthenia Gravis (26 words).

We are herewith proposing a shorter title, free of abbreviations:

Immunological and Structural Characterization of Titin Main Immunogenic Region; Domain I110 is the Target of Titin Antibodies in Myasthenia Gravis (20 words).

Point 3: The authors note that anti-titin antibodies play a role in the diagnosis of MG. Doe            s this also apply to the other autoantibodies?

Response 3: We have now clarified in the introduction that the initial MG diagnosis is achieved with the extracellular MG autoantigens (AChR etc.) (line 61), whereas among the intracellular autoantigens both titin and ryanodine receptor are biomarkers for MG subgroup classification (mostly for the differentiation of MG with or without thymoma) and treatment optimization (lines 66-68).

Point 4: It is not quite clear to me what the advantage is of clarifying the titin epitopes.

Response 4:     

  1. Some of the benefits of the present localization of the titin antigenic region include:
  2. i) it should lead to easier antigen production (being 1/3 of the MIR), resulting in less expensive diagnostic kits.
  3. ii) restricting the antigenic fragment may lead to less non-specific binding, i.e. more specific diagnostics.

iii) restriction of the antigenic region may facilitate the research on the relation of anti-titin antibodies with thymoma, and on the role of thymoma in the induction of MG.

  1. Future fine epitope localization could prove very useful if it shows that the exact epitopes for thymoma and non-thymoma patients (although all being located in I110) differ from each other. As stated already, anti-titin antibodies are found in early onset MG with thymoma and in late-onset MG independent of thymoma presence; therefore, so far, measurement of anti-titin antibodies is very useful only for early-onset MG patients. If we eventually identify thymoma-specific epitopes/antibodies, these will be invaluable for thymoma diagnosis in all MG age groups.

These comments are now presented in the Discussion, lines 570-573 and 558-562

Point 5: In the introduction, the authors should also mention the connection between MG and thymoma. This is not clear and is certainly not obvious to every reader.

Response 5: The thymus is the prime site of autosensitization and autoimmunity to AChR in MG pathogenesis [29]. Approximately 70% of MG patients present thymic changes, including hyperplasia and thymoma; the latter can be found in about 20-25% of MG patients worldwide [30]. Thymoma, likely via the AChR it contains and the defective autoimmune regulator (AIRE) or HLA class II expression, is believed to be involved in the induction of MG in thymoma-MG. Being a tumor, thymoma presence must be identified, for thymectomy to follow [31]. (We have added this in lines 93-99, and added the new refs 29-31).

Point 6: It is not clear to me how one wants to further characterize the MG on the basis of the anti-titin-AK if there are no real differences between the individual sera. This rather speaks against the search for anti-titin-AK as a biomarker.

Response 6: As stated above, further fine epitope localization could prove very useful if it shows that the exact epitopes for thymoma and non-thymoma patients (although all being located in I110) differ from each other. As stated already, anti-titin antibodies are found in early onset MG with thymoma and in late-onset MG independent of thymoma presence; therefore, so far, measurement of anti-titin antibodies is very useful only for early-onset MG patients (onset <40 years old). If we eventually identify the thymoma-specific epitopes/antibodies, these will be invaluable for thymoma diagnosis in all MG age groups. (See Discussion, lines 570-573).

We thank the reviewer for the instructive comments.

Round 2

Reviewer 3 Report

No further comments.